# Enhancing Existing Formal Home Care to Improve and Maintain Functional Status in Older Adults: Results of a Feasibility Study on the Implementation of Care to Move (CTM) in an Irish Healthcare Setting

**DOI:** 10.3390/ijerph191811148

**Published:** 2022-09-06

**Authors:** Frances Horgan, Vanda Cummins, Dawn A. Skelton, Frank Doyle, Maria O’Sullivan, Rose Galvin, Elissa Burton, Jan Sorensen, Samira Barbara Jabakhanji, Bex Townley, Debbie Rooney, Gill Jackson, Lisa Murphy, Lauren Swan, Mary O’Neill, Austin Warters

**Affiliations:** 1School of Physiotherapy, Royal College of Surgeons in Ireland (RCSI), University of Medicine and Health Sciences, D02 YN77 Dublin, Ireland; 2Primary Care Physiotherapy Services CHO9, Health Service Executive, D09 C8P5 Dublin, Ireland; 3Research Centre for Health (ReaCH), School of Health and Life Sciences, Glasgow Caledonian University, Glasgow G4 0BA, UK; 4Department of Health Psychology, RCSI University of Medicine and Health Sciences, D02 YN77 Dublin, Ireland; 5Department of Clinical Medicine, Trinity College, D02 PN40 Dublin, Ireland; 6Ageing Research Centre, Health Research Institute, School of Allied Health, University of Limerick (UL), V94 T9PX Limerick, Ireland; 7School of Allied Health, Curtin University, Bentley, WA 6102, Australia; 8enAble Institute, Curtin University, Bentley, WA 6102, Australia; 9Healthcare Outcomes Research Centre (HORC), RCSI University of Medicine and Health Sciences, D02 YN77 Dublin, Ireland; 10Later Life Training, Killin, Scotland FK21 8UT, UK; 11North Dublin Home Care (NDHC), D03 A6Y0 Dublin, Ireland; 12Graduate School of Healthcare Management, RCSI University of Medicine and Health Sciences, D02 YN77 Dublin, Ireland; 13Older Person Services CHO9, Health Service Executive (HSE), D09 C8P5 Dublin, Ireland

**Keywords:** feasibility, older adult, home care, community-dwelling, physical activity, intervention, care staff, training

## Abstract

**Background**: Care to Move (CTM) provides a series of consistent ‘movement prompts’ to embed into existing movements of daily living. We explored the feasibility of incorporating CTM approaches in home care settings. **Methods**: Feasibility study of the CTM approach in older adults receiving home care. Recruitment, retention and attrition (three time points), adherence, costs to deliver and data loss analyzed and differentiated pre and post the COVID-19 pandemic. Secondary outcomes, including functional status, physical activity, balance confidence, quality of life, cost to implement CTM. **Results**: Fifty-five home care clients (69.6% of eligible sample) participated. Twenty were unable to start due to COVID-19 disruptions and health issues, leaving 35 clients recruited, mostly women (85.7%), mean age 82.8 years. COVID-19 disruption impacted on the study, there was 60% retention to T2 assessments (8-weeks) and 13 of 35 (37.1%) completed T3 assessments (6-months). There were improvements with small to medium effect sizes in quality of life, physical function, balance confidence and self-efficacy. Managers were supportive of the roll-out of CTM. The implementation cost was estimated at EUR 280 per carer and annual running costs at EUR 75 per carer. **Conclusion**: Embedding CTM within home support services is acceptable and feasible. Data gathered can power a definitive trial.

## 1. Introduction

Ireland has a growing ageing population and those living beyond 80 years are set to grow by 94% between 2015 and 2030 [1]. Like many other countries, Ireland aims to enable people to live at home for as long as possible. Home care is healthcare or supportive care provided by a professional caregiver in the individual home where the patient or client is living, as opposed to care provided in group accommodations like clinics or nursing homes [2]. In Ireland, most formal home care is state-funded [3]. In 2021, an estimated 23.9 million hours of government-funded home support services were delivered to over 55,000 people aged 65 years and older in Ireland [4]. Many older people receiving home care are frail and at higher risk of adverse outcomes and increasing care needs over time [5,6]. Frailty is a distinctive health state related to the ageing process in which multiple body systems gradually lose their in-built reserves [7]. A systematic review and meta-analysis of physical activity programs for people receiving home care recommended the need for more evidenced-based trials tailored to this population [8]. Home care clients have expressed a preference for integrated lifestyle exercise and being physically active through activities they enjoy, rather than more structured exercise programs [9].

Through Care to Move (CTM) [10], older adults receiving home care are encouraged to set goals to do more ‘movement’ associated with self-care activities and, with support, integrate some strength and balance activities into their day as opposed to a prescribed set of exercises conducted for a set amount of time [11]. Trained home care staff support clients by using prompts to promote the movements that they can do at home when carers are with them, or when clients are at home alone, and these behavior change techniques are highlighted as potentially important for effective re-ablement [12]. CTM is best implemented across the whole workforce, because its primary aim is to bring consistency (of language, key messages) to teams delivering packages of care or those regularly engaging with older people in their homes. CTM approaches have not been researched and it is not known if it is feasible or acceptable to older adults receiving care or the home care staff who would provide these CTM approaches.

The aim of this study was to investigate the feasibility and acceptability of implementing CTM approaches to older adults living in the community who are receiving home care in Ireland. Our secondary outcomes explored physical function, physical activity, activities of daily living, balance confidence and health-related quality of life of home care recipients at multiple time points. We also estimated costs of implementing and running CTM. The feasibility and acceptability of the CTM approach from the point of views of managers within a home care service were sought. Feasibility and acceptability of the CTM approach by home care staff [13], and home care clients’ experiences of embedding physical activity prompts into their care [14], have been published.

## 2. Materials and Methods

### 2.1. Study Design

This feasibility study applied Bowen’s framework for feasibility studies [15] and the CONSORT extension for pilot and feasibility studies, MRC framework [16,17]. The study protocol has been published [11].

### 2.2. Participants and Recruitment

Home care clients were recruited from a not-for-profit home care provider. Care managers, supervisory staff and home care workers reviewed and screened a list of all service users based on the following inclusion and exclusion criteria. Inclusion Criteria: aged 65 years or older, had a Clinical Frailty Score [18] of 6 or less, had fallen at least once in the last year, received ≤5 h of home care a week and were independently mobile (with or without a walking aid). Exclusion Criteria: moderate to severe cognitive impairment, any unstable clinical conditions, receiving end-of-life care, or unable to safely follow instructions about exercising, moving or being more physically active. Participant recruitment took place from May 2019 to November 2020, which spanned COVID-19 public health restrictions.

### 2.3. Procedures

Home care staff, supervisors and managers received training in the delivery of CTM [10] between March 2019 and February 2020. Eligible home care clients were identified by a supervisor in the home care provider. They were provided with a study invitation letter, participant information sheet and received a follow-up call within a week to establish if they would like to participate. Following identification of suitable participants, a face-to-face meeting was arranged with the research physiotherapist to discuss the project. Participants were given up to seven days between receipt of the study information and being requested to give written informed consent to participate. Withdrawal from the study did not impact on the level of home care received. Subsequent home visits were arranged and completed by a research physiotherapist (VC) to obtain written consent and to provide any further study information. Participants were assigned to and then received CTM from a trained home care worker for 6 months. Assessments were due to be completed by a research physiotherapist (VC) during home visits at baseline, after 8 weeks of the intervention and at 6 months. Redeployment of the research physiotherapist for six months and continued social distancing restrictions on their return meant we had to offer telephone assessments instead of home visits. Because of the COVID-19 pandemic and changes in work patterns and staffing challenges the interviews took place between March and May 2021.

### 2.4. The Care to Move Approach

CTM provides a series of consistent ‘movement prompts’ to use and embed into existing movements of daily living. It offers a series of key messages for home care staff to communicate during all interactions about sitting and moving more with a view to encouraging and empowering older people to make different decisions in the longer term to better contribute to their health, well-being, confidence and independence. CTM is not a structured exercise program, but is designed to increase movement and embed repeated movements into everyday life tasks. Specifically, CTM seeks to achieve this by giving home care staff confidence to have empowering and motivating interactions with clients. The detailed CTM intervention description is published [11]. During the delivery of training in CTM to the home care staff and key trainers, the CTM interventions was mapped against behavior change techniques from the COM-B Taxonomy [19]. The CTM training for support workers has three key themes/approaches to allow this ‘embedding’: Communication skills to have purposeful conversations about movement (providing a structured framework); a series of targeted, specific movements and prompts (for key movements already being performed as part of the usual package of care, daily living); where applicable, motivating and empowering older people to carry out home exercise programs prescribed by therapy services. The specific movements and prompts included ‘prepare to move’ prompts (hip and buttock movement towards the front of the chair before a rise, foot placement and powering upright with some foot pedals to aid circulation and reduce chances of postural hypotension); prompts to improve ‘ADLs’ (heel raises and knee bends to get things in/out of cupboards, balance tasks waiting for kettle to boil near a solid fixed support); reviewing successes and movements since last visit or during visit. The behavior change techniques (BCTs) covered in the CTM training with home care staff, and those used in the delivery to the clients, have been described [11] but were also mapped during the initial training of home care staff by two authors (DAS, FD) (see Appendix A). Over 3 visits during the initial 8 weeks of CTM delivery, the research physiotherapist worked with the CTM-trained home care staff and home care client to demonstrate how the movements could be completed safely and effectively. Beyond the collaboration with the physiotherapist, home care staff were key to delivering CTM, by consistently motivating and encouraging participants. For 6 months, CTM-trained home care staff integrated CTM in their regular home care visits (at least once per week) for home care clients. The CTM participants were encouraged and prompted to undertake movements specifically aimed at improving posture, circulation, mobility, balance and strength alongside daily activities. These movements could be done several times during the day or were prompted when the home care staff visited. For this feasibility study, home care staff asked participants to complete a weekly report in which they monitored falls or major health changes and healthcare use. Home care clients were also given a calendar to tick each day they performed an activity during the study period.

### 2.5. Primary Outcomes

We aimed to recruit 40 home care clients with the goal of retaining 30 to follow-up. The primary outcomes were: number of home care clients (participants) recruited (recruitment); number of participants that provided data at 8 weeks and 6 month follow-up (retention); number of participants that showed engagement with CTM and progression over time using care documentation (adherence); data loss in questionnaires and secondary outcome measures; any adverse events, falls, major changes in health and healthcare use, related or not to the CTM intervention, collected in a weekly report by participants and given to the carers. Due to the potential for multiple adverse events (e.g., health deterioration and hospitalizations) in this population, we chose to report only falls as adverse events. This choice was to examine the potential of an increased risk of falls by exposure to risk through more movement. It was not considered feasible, in terms of added paperwork burden to a busy service, to have a specific adverse event report form and so falls were reported through weekly reports and calendars from clients and usual care reporting and documentation.

### 2.6. Secondary Outcomes

We initially planned for the research physiotherapist to record study outcomes at baseline, after 8 weeks and at 6 months [11]. However, considerable disruptions to the study logistics occurred due to the COVID-19 pandemic, so the final follow-up visit was scheduled whenever possible.

The following secondary outcomes were assessed: Timed Up and Go (TUG) test [20]; Lower body strength (30-s chair stand test) [21]; Nottingham Extended Activities of Daily Living (NEADL) scale [22]; Balance confidence (10-item Activity-specific Balance Confidence (ABC) scale [23]; CONFBal (10-item falls self-efficacy scale) [24]; Self-reported physical activity (PhoneFITT) [25]; Quality of life (SF-36 and EQ-5D-5L questionnaires) [26,27]; Exercise Self-efficacy Scale, Social Cognitive Theory Scale (adapted from [28], see Appendix A); Barriers to exercise, reduced from 7 questions to 4 and ‘exercise’ changed to ‘move more’ (adapted from [29], see Appendix A), Outcomes Expectations Scale (adapted from [30], see Appendix A) and Intention and planning Scale (adapted from [31], see Appendix A). Additionally, we examined delivery (fidelity) and compliance to CTM and explored the capacity to deliver CTM with a home care provider.

### 2.7. Data Analysis

This study was a feasibility study, so no formal power calculation was conducted. Descriptive statistics of secondary outcomes were performed, namely reporting *n* at different time points, and mean, standard deviation, median and interquartile range. Effect sizes were calculated using Cohens d_s_ [32], for each outcome measure (between T1 and T2 and between T1 and T3) to give an indication of effect for future studies. Participants’ individual identifiers were used as level 1 and time was used as level 2 factor within which measurements were nested. Accordingly, the models take account of within-person correlation of measurements, that is where several measurements of one participant over time are potentially correlated. Models were adjusted for participant age (mean-centred) as this was expected to influence the outcome variables. Analysis was conducted using Stata version 17 (College Station, TX, USA). Interviews with managers were recorded, transcribed and analyzed using thematic analysis and coded for acceptability and feasibility [33].

### 2.8. Acceptability and Feasibility Outcomes

Acceptability was evaluated by: Number of home care clients (participants) that were recruited (recruitment); number of participants that showed engagement with CTM and progression over time using care documentation (adherence); perspectives of home care managers, home care staff and home care clients. Feasibility was evaluated by: number of participants that provide data at 8 weeks and 6 month follow-up (retention/attrition); data loss in the questionnaires and tests within the secondary outcome measures, and time taken to complete; any adverse events—falls, related or not to the CTM intervention, and perceptions of home care managers, home care staff and home care clients.

### 2.9. Health Economic Evaluation

We examined the implementation and running cost of CTM from the home care provider’s perspective and extrapolated costs to the full home care sector in Ireland. Implementation cost of CTM included home care staff time for the initial training, trainer fees and modest investments in the IT equipment/software to integrate CTM in the existing system. Running costs related to briefing and training maintenance for staff, using key trainers within the organization, and additional staff time to introduce CTM to new clients. No CTM training cost was considered for new staff as we assumed that CTM will be an integral part of the general orientation of new staff. Hourly staff salaries and unit costs were received from the home care provider manager (DR) and physiotherapist (VC) (2021-price level). Only incremental costs of CTM were considered, as CTM will not affect the level of general home care delivered to clients. No discounting was applied to the yearly costs assessment. In order to estimate the cost implications related to staff, we assumed that the home care provider employed 150 home care staff. We also calculated the additional cost of CTM per client with the assumption that the target group included 350 clients and the implementation cost were depreciated over 3 years. We used two approaches to extrapolate the costs to the Irish population based on an average CTM cost per client or per home care staff. The exact carer numbers are unknown (Walsh 2018). In 2021 the national home support service provided 20.26 million support hours [4]. This volume of home support service would be delivered by 15–20,000 home care staff (The HSE service plan 2021 states that the home support will deliver 24.26m hours in 2021 (Older Persons’ Services, Home support, page 54). Assuming each home care staff member delivers between 1620 and 1250 h/year, there will be between 14,975 or 19,408 staff employed). Assuming that each home care staff member supports 10 clients per year and that 20% are in the target group for CTM, we assumed that the national CTM population will include 30–40,000 clients. We assume that 15% will be new clients who require introduction to CTM.

### 2.10. Ethical Approval and Trial Registration

Ethical approval was granted by the Royal College of Surgeons in Ireland Research Ethics Committee (REC -2018:1489). Clinical Trial Number: ISRCTN72605421.

## 3. Results

### 3.1. Training of Home Care Staff in CTM Approach

A total of 53 home care staff, supervisors and managers received training in the delivery of CTM, 15 in the initial training by LLT (2 home care managers, 2 home care supervisors and 12 home care staff) [10] and 38 (3 supervisors and 35 home care staff) over 4 courses led by a Key Trainer (VC). The Key Trainer was unable to do any more CTM training with carers after March 2020 as she was redeployed from her research post to frontline therapy services. Following training and into the COVID-19 pandemic, the home care provider lost 23 CTM-trained carers (left/retired 18; maternity leave 1; sick leave 1; reduced hours 1; promoted to supervisor 2) which left only 30 CTM home care staff able to provide the CTM approach with home care clients.

### 3.2. Feasibility Outcomes

#### 3.2.1. Recruitment

One hundred and fifty-one home care clients were first identified by the home care team across the feasibility study trial duration. Forty-six were identified pre-COVID restrictions and 105 after March 2020, when a lockdown for nearly 3 months occurred in Ireland. Even when this lockdown was lifted, restrictions on social distancing and time spent with people in close proximity continued until February 2022.

Pre-COVID lockdown: Of the 46 clients identified from May 2019 to March 2020 (9 months), 1 was excluded (2.2%), leaving 45 eligible and approached, of which 4 declined (8.7%) (Figure 1).

Post-COVID lockdown: A further 105 were identified from March 2020 to November 2020 (8 months), mostly by the home care staff. There was a noticeable difference in percentage eligible after March 2020, when the supervisors were less involved in recruitment. Of those identified, 71 were excluded (67.6%) because 12 (16.9%) were deemed unsuitable by the supervisor, 28 (35.2%) had cognitive impairment or language barriers which precluded their engagement, 4 (5.6%) were unable to mobilize out of their wheelchair, 4 went into hospital (5.6%), 1 moved away from the area, and 11 (15.5%) were in pain or too unwell to be involved. Of note is that further 11 (15.5%) had to be excluded as they no longer had a home care support worker that was trained in CTM. This left 34 approached, of which 20 (58.8%) declined. Although no reasons were sought for people declining, many mentioned being uncomfortable in getting too close to the home care support staff and may have felt that being involved in a research study during uncertain times (because of the pandemic) was not high on their list of priorities.

In total, 79 (52.3%) of all home care clients screened for inclusion were eligible and approached for inclusion.

Fifty-five home care clients (69.6% of total eligible) consented to participate, but 4 withdrew consent, 1 was hospitalized before T1 and 16 wanted to wait until COVID-19 had no social restrictions. During this time, 6 clients cancelled their home care packages and other clients did not end up taking part in T1 assessments (Figure 1). The majority of included home care clients were female (85.7%), the mean age was 82.8 (sd 7.84) years (median 84, range 60–96 years). Marital and living status, use of walking aids and Rockwood frailty score are presented in Table 1.

#### 3.2.2. Adherence

COVID-19 had a major impact on the adherence of the home care staff to delivering the CTM approach within clients’ homes. For 3 months of lockdown and for some time after, care plans were modified and priority care plans activated by the home care provider for safety and well-being of staff and clients. Twelve of 35 (34%) clients showed engagement with CTM and progression over time using care documentation (adherence). Due to COVID-19 workforce challenges, the research physiotherapist was redeployed away from the research to frontline practice from March to September 2020, so no assessments could be arranged. From this point, assessments for T2 became remote in nature, by telephone or by mobile phone, where the home care staff showed the physiotherapist a client performing the secondary outcomes using a mobile device. Clients’ adherence to completion of the weekly report monitoring falls or major health changes, and the calendar to tick each day they did additional activity, was very poor, particularly as clients were not prompted from the start of social restrictions. After 2 months, these were discontinued and reliance on home care documentation and reports from home care staff were used to record adverse events. We had planned to examine fidelity to the delivery of the CTM approach by examining CTM care documentation and observations, however, observations were not possible with the start of the pandemic and CTM-specific paperwork was not completed by many of the staff in the ensuing home visits to save time.

#### 3.2.3. Retention

Retention was severely affected by COVID-19 social restrictions and ill-health. We aimed to recruit 40 home care clients with the goal of retaining 30 (75%) to follow-up. At T1, we completed baseline assessments on 35 clients. At 8 week follow-up, we retained 60% (*n* = 21) clients and by T3, at around 6 months, we retained 37.1% (*n* = 13) of home care clients. The main reasons for drop-out was ill-health and cognitive decline, and many follow-up remote assessments were not possible. Two participants no longer had CTM-trained home care staff as their home care support. Six clients cancelled their home care packages.

#### 3.2.4. Data Loss in Questionnaires and Tests; Time Taken

Missing or incomplete data (Table 2) ranged from 0% to 38.5% (for T3 Exercise self-efficacy), depending on the outcome measure. Exercise self-efficacy had the highest level of missing data as it was the last questionnaire assessed as part of the battery of measures. Data for the Rockwood Frailty Scale, PhoneFITT and SF-36 are available in Appendix A. Face-to-face assessment of secondary outcomes took approximately 60–90 min and slightly shorter when done by telephone. For some participants, the assessment had to be done over two home contacts due to participant fatigue.

#### 3.2.5. Adverse Events

There were no adverse events associated with the CTM approach or movements. Over the course of the CTM approach, 23 falls were reported, with 11 requiring ambulance call-outs and 2 fractures (knee and arm). No falls were reported as being related to the CTM approach.

#### 3.2.6. Perceptions of Home Care Managers; Acceptability

Four managers were interviewed between March and May 2021. Interviews lasted 60 min on average (range: 55–75 min). Exemplar quotes supporting the findings, described next, are presented in the Appendix A. Overall, managers were very positive and supportive of CTM as a whole-workforce approach to care. All agreed that the concept of CTM is simple and represents a shift away from a more traditional task-orientated approach to caring for older people at home. At a commissioning/strategic level, managers indicated that CTM enhanced the package of care offered to older people in the community and had a particular role in preventing and managing frailty at home. Moreover, CTM had the potential to enable independent living as clients are visited in the privacy of their home. It was also felt that CTM was adaptable for everyone whose home was the established environment of care, maximising their involvement in daily activities in their own homes.

Managers indicated that CTM was empowering for clients since the home care staff and client were working together as a unit, where relationships of trust are built over time. Moreover, home care workers were viewed as key players for the successful delivery of CTM within a supportive framework; thus CTM created an opportunity for managers to further explore the role of home care workers and support them in developing new skills. In this regard, managers acknowledged the preparatory work that was required to introduce and embed CTM into existing services. This included education and training of home care workers, supervisors and managers about CTM and how it could be integrated into the existing care plans of clients in their caseload, for example both clients and families were encouraged to engage with CTM to maximize its potential and foster behavioral change. The training needs also included introducing CTM as an approach to care with a focus on understanding movement as distinct from exercise. The potential of integrating CTM in home care in the future was well recognized along with the value it added to home visits during the study period.

#### 3.2.7. Perceptions and Views of the CTM Approach Implementation by Home Care Managers: Feasibility

Because of the impact of the COVID-19 pandemic, the discussion around feasibility included many views about whether CTM could be embedded when staffing levels were seriously affected and social distancing was paramount. In addition to the need for training in CTM, managers recognized that home care workers needed the support of a CTM-trained supervisor to deliver and integrate CTM in home visits. Many managers viewed a leadership role as crucial for embedding CTM in home care and developing an awareness of how small changes can make a big difference to the client. Additionally, it established the purpose of CTM with the client and at the same time supported the home care staff in delivering CTM. The feedback loop to the supervisor was seen as essential to support home care staff in addressing any challenges which may emerge when visiting a client.

Managers also mentioned unavoidable challenges that arose when introducing CTM. Home care workers are lone workers who operate in a reactive and fast paced service where priorities of care are set in advance, tasks need to be completed and the time for each visit is logged. Consequently, CTM may have been perceived as an expanding role by some home care staff while others felt they were already encouraging movement during their normal visits. Managers also acknowledged that initiating CTM conversations may be challenging for some home care workers and that more training was required to tease out the subtle difference between physical activity, movement, and exercise and ensure home care staff reflected on the words they used to prompt movement.

Furthermore, the rearrangement of services when COVID-19 began posed a challenge when preparing and implementing CTM. Despite these challenges, the managers pointed out that home visits continued during the COVID-19 lockdowns, and that maintaining the safety of clients and carers was paramount.

An unintended consequence of CTM was the initiation of a change management process. At an organizational level, managers suggested that CTM training could be aligned to existing training programs, such as manual handing and staff mentoring. Managers recommended that a CTM module should be included in the Further Education and Training Awards Council of Ireland (FETAC) level 5 accredited training program for home care workers. The CTM approach also opened debate to re-examine the role of home care workers and maximize their capabilities and contribution to the care for older people, which may be overlooked in the broader context of community care. Additionally, the importance of partnering with the client and their family was recognized, as these have key roles in fostering the behavioral change required to embed CTM in daily living. As a result of the evaluation of the acceptability and outcomes of this study, we updated the Logic Model presented in the protocol paper [11] (Figure 2).

### 3.3. Secondary Outcomes

Descriptive statistics for secondary outcome measures, from baseline to T2 and T3, along with effect sizes for change are presented in Table 2. Analysis of the secondary outcomes of PhoneFITT and SF-36 domains are presented in Appendix A.

There were medium effect sizes seen with increased exercise self-efficacy and EQ-5D-5L at 8 weeks and 6 months (Table 2 and Appendix A). Smaller effect sizes were seen with signs of improvement in TUG at 8 weeks but not 6 months, 30 s chair stand at 8 weeks and 6 months, improved confidence in maintaining balance, and Activity-specific Balance Confidence (ABC) Scale, NEADL at 8 weeks and 6 months (Table 2). EQ-VAS showed a small effect size improvement at T2 which disappeared by T3. Medium effect sizes were seen with increased recreational physical activity on the Phone-FITT at 8 weeks and 6 months (Appendix A). However, NEADL scores declined over the course of the study and with lockdown (T3) (small effect size). Quality of life using the SF-36 domains showed that physical functioning, role limitations due to emotional difficulties and social functioning all increased significantly from T1 to T2 but then in some instances dropped again at T3. (Appendix A).

The proportion of clients who felt that lack of company or lack of interest were barriers to them being physically active, reduced over the study period (Appendix A). Lack of company as a barrier was reported by nearly 60% at TI and only 40% at T3. Reduction in barriers were seen at 8 weeks for both poor health (which rose again in the pandemic) and lack of interest (which stayed low).

In terms of Self-evaluation Outcome Expectations, completion rates of this questionnaire were poor at T2 (16 people) and T3 (*n* = 10 people). A total of 70.6% recorded agreement or strong agreement that moving more helped manage stress, mood and gave a sense of accomplishment at baseline and this rose to 83.3% at T2 and remained at that level at T3 (Appendix A). Social Expectations rose from 61.8% (T1) to 75% (T2) agreeing that moving more would help them socialize and remained at 70% at T3 despite the social restrictions posed by the pandemic. In terms of Physical Outcomes Expectations (moving more will improve ability to perform ADLs, overall body functioning, help bone and muscle strength) 80.9% agreed or strongly agreed at baseline and this rose to 90.6% at T2 but had dropped to baseline levels by T3. Completion rates of the questionnaire on intention and planning was also poor at T2 and T3, there were noticeable positive changes in reported intention and planning activities to move more (Appendix A). Figure 3 shows that in all aspects of intention and planning to move more, there were improvements from baseline to T2 at 8 weeks. There was some reduction by 6 months, strongly influenced by the pandemic as most T3 measures were taken mid-pandemic (Figure 3).

### 3.4. Cost Analysis of CTM Implementation

Embedding CTM in home care permanently will occur various once-off implementation costs as well as annual running costs. Overall, we estimate that cost of full implementation of the CTM program into the home care support organization at EUR 42,100. The annual running cost of CTM was estimated at EUR 11,000. The implementation cost includes the fees and salaries paid to provide the initial training for 20 participants including future ‘key’ trainers, CTM leads, care providers, supervisors and management representatives. The two external trainers who were responsible for planning and conducting the two-day training program and subsequent consultation and support received a fee of EUR 4600. The subsequent training of the remaining 130 home care staff was conducted over two days by the ‘key’ trainer in smaller groups with 10 participants. The cost of these 2-day training sessions included salary to the trainer (EUR 3500), materials (EUR 3100), in total EUR 8000. Salaries for 150 participating staff members assumed at EUR 190 per person equates to EUR 28,500. In addition, the once-off cost adapting CTM into the IT system required a cost of EUR 1000. The total implementation cost for the home care provider with 150 staff was thus estimated at EUR 42,100. This would equate to an investment of EUR 280 per staff member or EUR 140 per client.

The annual running cost would include staff time related to client introduction to CTM. This would include one hour with a home care staff and the CTM lead corresponding to the salary cost of EUR 40 per client or EUR 2000 for 50 new clients. Each staff member would attend quarterly briefing sessions and a biannual renewal session with a total salary cost of EUR 9000 per year. The estimated annual running cost at EUR 11,000 equates to EUR 75 per staff member or EUR 35 per client.

These feasibility study cost estimates for one home care provider can be used to extrapolate the cost of implementing CTM throughout Ireland. With an assumed home care staff number between 15–20,000, the implementation costs can be estimated to be between EUR 4.2 and EUR 5.6 million as a once-off investment and the corresponding annual running costs will be between EUR 1.1 and EUR 1.5 million.

## 4. Discussion

This research aimed to evaluate the feasibility and acceptability of the CTM approach being embedded to increase physical activity, thereby enabling older adults supported by a home care service in Ireland, to maintain independence. CTM aims to be a sustainable approach to ageing in place through enhancing home care services with embedding daily movement prompts in the care regimen. This feasibility study of CTM includes data from 35 home care clients who were predominantly female (85.7%), had an average age of nearly 83 years and were predominantly frail. We had aimed to recruit 40 clients and retain 75% to follow up. However, during the course of this feasibility study, the COVID-19 pandemic led to a lockdown and then strict social distancing and avoidance of contact with vulnerable older people. This led to difficulties in training home care staff in CTM, recruitment of home care clients, loss of trained home care staff from the workforce, redeployment of the research physiotherapist to frontline duties leading to delays in T2 and T3 assessments and many older home care clients declining in both physical and mental health. This decline in physical function and in mental health has been shown to be considerable even in older adults who live independently without a care package in Ireland [34]. As a result, we did not meet our target of recruiting 40 home care clients and retaining 30 to follow up at 6 months, indeed we retained only 37.1%.

We delivered CTM training to 35.3% of the home care workers within a home care provider before the lockdown but we could not continue training more because of social restrictions and workforce issues. This meant some home care clients could not be enrolled as they did not have a CTM trained home care worker in order to embed the approach. During the study duration, the home care company lost 43.3% of the CTM-trained home care staff, which also affected adherence to the approach with some clients who no longer had a trained home care staff worker to prompt them. We saw a high conversion of those screened by supervisors prior to COVID-19 converting to eligible and consenting participants, but this dropped considerably when supervisors were unable, during the pandemic, to spend time on the screening of participants and instead the home care workers recommended potential participants.

However, throughout the challenging recruitment period, home care clients were unlikely to decline the offer of CTM and of the 13 who completed assessments at T3, 12 showed good engagement with CTM and progression over time from the care documentation, suggesting it is an acceptable intervention. The interviews with the managers confirmed the CTM approach was acceptable, empowering, and (in a world without a pandemic) feasible, with the potential to prevent and manage frailty for those receiving care at home. Managers were very supportive of CTM, perceived it as beneficial both for home care clients and carer motivation, and suggested that CTM should be embedded in home care across the country. Managers described CTM as a whole-workforce approach to caring for older people at home that builds on partnership between the carer, home care recipient and their families. Certainly, the CTM approach was welcomed and supported by home care workers and clients alike [13,14]. There were no adverse events as a result of movement behaviors encouraged by CTM, however CTM is not suitable for those who require carer support to stand out of a wheelchair or who have moderate to severe dementia, so this aspect of eligibility should be retained.

Specifically, in line with previous research [35], managers highlighted the need for training and education for successful CTM integration, e.g., to address the misunderstanding of movement, as opposed to exercise. Two days of CTM training appears feasible; the same amount of training was provided during the implementation of a successful re-ablement service in Edinburgh [36]. Again, as has been seen before [35], Managers acknowledged the importance of supervisor support within the home care service to facilitate integration of movement prompts into the care plan. Additionally, in line with prior research [36], managers identified the role of families to support behavior change in the older adult.

A perceived barrier to effective service delivery is the nature of home care, which is characterized by pre-set responsibilities which individual carers have to respond to within short time windows. However, the evaluation of a Scottish re-ablement service found a 46%-reduction of required care hours among community-referred home care clients within a 6-week period, while traditional care clients experienced increasing care requirements (+14%). Accordingly, the authors conclude that the initial time investment in re-ablement would free up timely capacity among home care services [36]. In our study, managers further mentioned interruption to care delivery due to COVID-19; however, they perceived that home care and CTM were able to be delivered safely despite this challenge.

In order to implement CTM nationally, managers see the need for system-wide change and infrastructure development, which includes IT and care planning processes, including within individual home care services. Interviews also showed that a CTM module would be suitable for inclusion in the Further Education and Training Awards Council of Ireland (FETAC) level 5 accredited training program for home care workers [37]. Barriers to its implementation include the necessary changes to paperwork, potential to affect caseload, and training time (although valued) for the supervisors and home care workers.

Data loss was minimal (in those that were assessed) but perhaps over-burdensome as some clients got very tired, and the change of protocol to gather assessment data by phone, rather than face to face, made some questionnaires and tests more challenging to complete.

Although not powered to show effectiveness, we saw some changes in secondary outcome measures that suggest CTM has the potential to help prevent or manage frailty and improve quality of life. This sample of home care clients had very low activity specific balance confidence (<50%) at baseline comparable to previous studies of older adults receiving home care [38] and older adults with a history of falls [39]. Yet, there were promising improvements seen in both activity-specific balance confidence and confidence in maintaining balance scales at T2 and T3. Promising but small effect sizes were seen in Timed up and go and 30 s chair stand, which could be used to power a future effectiveness study. Importantly, home care clients increased in self-efficacy for exercise, showed reduced barriers to movement and showed improvements in intention and action planning for moving more. We did not see an effect on physical activity. Indeed, physical activity scores in the home care clients were considerably lower than those seen in a group of older veterans living in Canada [40]. The Phone-FITT Household FD score for our sample at baseline was 11.1, compared to 20.7 in the veterans, who had informal caregivers (spouses, etc.). This is perhaps to be expected as our sample received formal care in the home. Phone-FITT Recreational FD scores were 10.2 in our sample and 14.7 in the veterans. CTM aims to increase physical activity in the home environment, however, using the Phone-FITT, only 5 of the 14 questions could have been influenced by CTM (changes in light housework; making meals or cleaning dishes; exercises to strengthen legs; stretching or balance exercises; walking for exercise), so for a future study, a different method of recording physical activity should be sought, or perhaps only a sub-set of questions be asked.

The analysis of costs showed that initial implementation of CTM is relatively inexpensive compared to its annual running costs, amounting to a once-off investment at EUR 42,100 and annual running costs at EUR 11,000 in a home-care provider with 150 care staff. The training costs of EUR 280 per home care staff member is reasonably similar to the training costs (GBP 191) in a Scottish re-ablement service. However, the running costs within the Scottish service were much higher than estimated here (GBP 240 per client for 6 weeks, compared to EUR 35 annually in our study). The higher costs in the Scottish analysis is mainly due to high administrative and managerial costs and the inclusion of occupational therapists. In both studies, the care costs remained unchanged in the re-ablement clients compared to traditional home care clients [36]. If CTM could be implemented at national level, the once-off implementation cost will range between EUR 4.2 and EUR 5.6 million, and annual running costs between EUR 1.1 and EUR 1.5 million. These cost estimates are sensitive to the assumptions of the staff and client numbers. The provider that participated in this study has a client-to-staff ratio of 10:1 compared to the national ratio of 10:6. Throughout Ireland, there is a shortage of home care staff and frequent turn-around. This might be influenced by staff job satisfaction. The CTM approach appears to increase job satisfaction and could positively influence the turn-around of care staff. This is consistent with the Scottish evaluation of re-ablement services that showed a reduction in the workload of carers [35]. This implies that the assumptions related to the lower cost estimate are unrealistic and that the investment is more likely to be at the higher cost range. The train-the-trainer concept lowers the cost of initial training. This enabled the 150 home carers to be trained at a much lower cost than if all carers should be trained as the first cohort of 20 carers through Later Life Training (EUR 3600). Initiation of this train-the-trainer principle enabled the cost of the in-house training to be costed as hourly salary rather than external consultant fees.

To our knowledge this is the first study to explore training costs associated with an aspect of care delivery within the homecare sector in Ireland. The recognized qualification for home support workers is a level-5 Quality and Qualifications Ireland (QQI) Award [41]. However, it is unclear who meets, or how the costs of the staff achieving this award are met. The importance of training was highlighted in a public consultation undertaken by the Department of Health in Ireland [42], similarly the national health and social care regulator (Health Information and Quality Authority) have highlighted training as being essential to good standards [43]. In 2022, a Cross-Departmental Strategic Workforce Advisory Group on Home Carers has been established and training will be a major focus of the work [44]. The work of this group may provide clarity on how the costs of training this workforce are met.

CTM is an approach that might benefit from being a fundamental part of existing training modules, rather than a separate distinct training program, thus the costs would be incorporated within these existing training costs. Other findings from this study highlighted the potential of developing CTM “champions” [13], this type of approach emerged from efforts to improve dementia care [45]. The costs of implementing and running CTM should be balanced by the positive outcomes. While this study lacks statistical power to show reliable improvements in health, the data analysis suggests potential improvements in physical function, confidence and health-related quality of life. It is noteworthy that we did not identify a deterioration in health status despite data being obtained during the COVID-19 period. Future research should similarly investigate the cost-effectiveness of the CTM approach in a larger sample of home care clients and consider longer-term outcomes of place of residence and proportion that move into care home settings.

This study’s strength is embedding CTM training through ‘key’ trainers into a sustainable model of training and although the first 15 home care staff were trained by Later Life Training, this included training a ‘key’ trainer within the team who continued to train an additional 38 home care staff, and these ‘key’ trainers could train the additional carers if CTM was scaled up. However, there are some limitations to the generalizability of the findings. This feasibility study was unavoidably impacted by COVID-19, for example, the research physiotherapist was seconded to frontline healthcare, making initial T1 visits and follow-up visits impossible within the original timeframes and leading to a smaller sample size at both recruitment and follow-up than planned. Some of the outcome measures had to be gathered by telephone, rather than face-to-face due to social restrictions in place during the pandemic. It is possible that this led to some outcome measures being less accurately reported. We had planned to look at fidelity of delivery of the CTM approach, but the pandemic meant we could not perform any observations and the care staff stayed in peoples’ homes for as short a time as possible and the CTM study paperwork was not high priority to complete. This will have to be considered in a future trial. However, the perspectives of managers and the encouraging changes to many outcome measures suggest that embedding physical activity initiatives within home support services is feasible and worthwhile and our results can inform a future definitive trial.

Our cost calculation only provides a direction, and exact budgetary requirements will need to be assessed in future research. We acknowledge that the number of people in need of home care is projected to increase in Ireland [1,3]; thus it is likely that more than 3% of existing home care clients newly are referred to the service in future years and that fewer clients drop out of the services. Furthermore, we calculate costs for the entire service (i.e., 150 carers), whereas only 54 carers were trained as part of this study. This extrapolation was necessary as training of the first 15 carers is much more expensive than subsequent trainings, and as training of the entire home care provider workforce would be required to maximize effective CTM delivery for all home care clients. Additionally, we did not have access to healthcare use data and we did not investigate outcomes in a control group; therefore we could not perform a full health economic evaluation.

## 5. Conclusions

This feasibility study of CTM approaches suggests that embedding physical activity initiatives within home support services is feasible, acceptable and managers, supervisors, home care staff and clients consider it worthwhile. Costs required for staff training and full CTM costs per home care recipient are low, and training schemes/design contribute to a sustainable extension of home care services. While challenges remain with regard to change management and workforce planning, the trends for improvement within the outcome measures and the positive views of managers suggests that CTM approaches should be further assessed in a larger definitive study.

## Figures and Tables

**Figure 1 ijerph-19-11148-f001:**
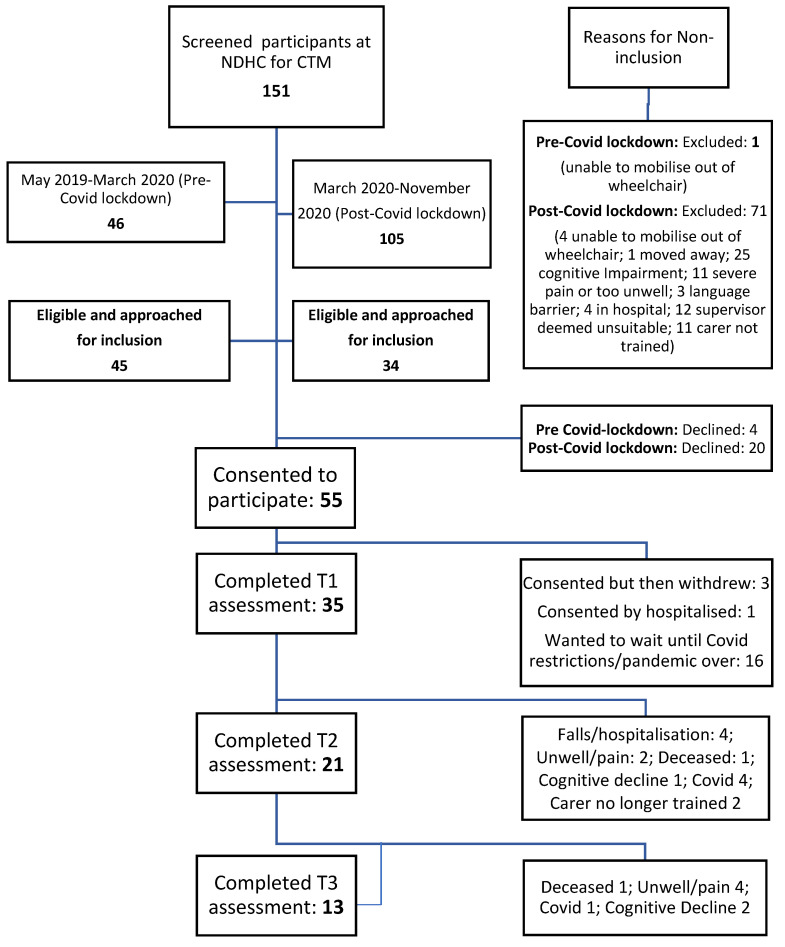
Flow of the participants in the CTM feasibility study (before and after COVID-19 social distancing restrictions).

**Figure 2 ijerph-19-11148-f002:**
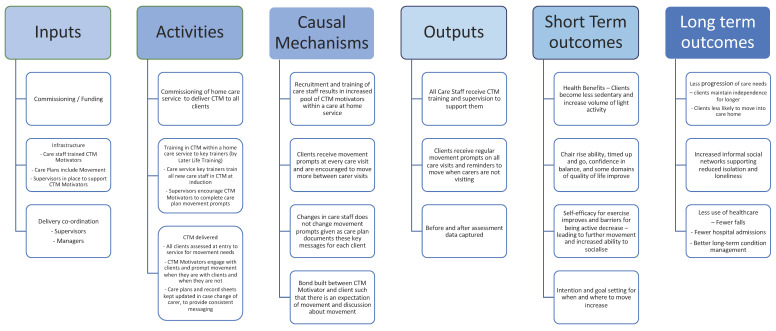
Logic model for CTM Implementation.

**Figure 3 ijerph-19-11148-f003:**
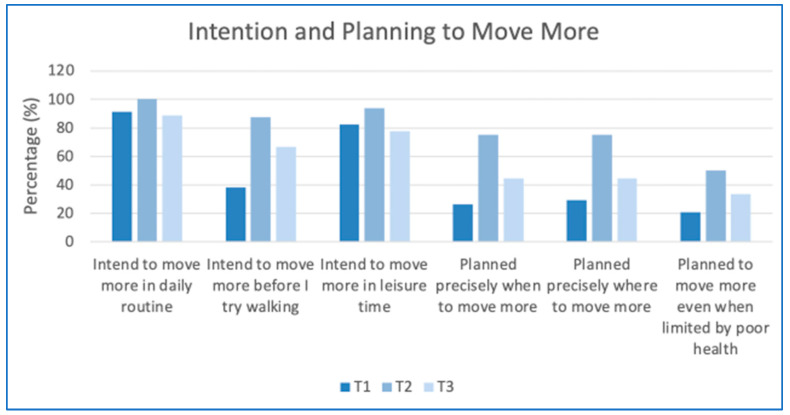
Intention and planning over the 6-month timeframe.

**Table 1 ijerph-19-11148-t001:** Participant characteristics at baseline.

Variable	Variable Description	N(%) out of Total 35
Gender	Male	5 (14.3%)
	Female	30 (85.7%)
Marital status	Married	9 (25.7%)
	Single	10 (28.6%)
	Widowed	16 (45.7%)
Living status	Lives alone	24 (68.6%)
	Lives with other	11 (31.4%)
Uses mobility aid	No aid	12 (34.3%)
	Stick	12 (34.3%)
	Frame ^1^	10 (28.6%)
	Wheelchair	1 (2.9%)
Rockwood Score	3	1 (2.9%)
	4	3 (8.6%)
	5	12 (34.3%)
	6	19 (54.2%)

^1^ Frames could be a 2, 3 or 4 wheeled frame, no wheels or a walking rollator.

**Table 2 ijerph-19-11148-t002:** Descriptive statistics and data loss for secondary outcome measures at Time 1, Time 2 and Time 3.

Outcome Measure	Time 1	Time 2	Time 3
	Baseline	8 Weeks	6 Months or Later
Timed Up and Go (secs) $: *n* (% of total)			
Mean (SD)	33 (94.3%)	20 (95.2%)	11 (84.6%)
Median (IQR)	21.8 (10.8)	19.1 (9.3)	22.8 (11.3)
Cohens d_s_	19.6 (10.9)	18.8 (11.0)	18.8 (11.0)
	_	−0.26 ^a^	0.1
30 s sit to stand (number) ¶: *n* (% of total)			
Mean (SD)	33 (94.3%)	18 (85.7%)	12 (92.3%)
Median (IQR)	7.0 (2.7)	7.8 (3.4)	8.0 (4.4)
Cohens d_s_	7.0 (4.0)	7.0 (5.5)	7.0 (4.5)
	-	0.28 ^a^	0.31 ^a^
Nottingham Extended ADL (score) ¶:			
*n* (% of total)			
Mean (SD)	35 (100%)	21 (100%)	13 (100%)
Median (IQR)	55.8 (13.0)	52.3 (11.5)	51.9 (9.3)
Cohens d_s_	57.0 (19.0)	57.0 (19.0)	53.0 (12.0)
	-	−0.28 ^a^	−0.32 ^a^
ConfBal (score) $: *n* (% of total)			
Mean (SD)	35 (100%)	20 (95.2%)	13 (100%)
Median (IQR)	21.3 (4.3)	19.3 (3.7)	20.2 (3.8)
Cohens d_s_	21.0 (6.0)	19.0 (3.3)	20.0 (5.0)
	-	−0.49 ^a^	−0.28 ^a^
ABC balance confidence (%) ¶: *n* (% of total)	34 (97.1%)	18 (85.7%)	12 (92.3%)
Mean (SD)	48.3 (18.5)	44.9 (13.7)	41.9 (19.5)
Median (IQR)	45.9 (16.9)	45.3 (23.0)	38.8 (30.3)
Cohens d_s_	-	−0.20 ^a^	−0.34 ^a^
EQ-5D-5L (%) ¶: *n* (% of total)	35 (100%)	20 (95.2%)	12 (92.3%)
Mean (SD)	0.549 (0.195)	0.693 (0.195)	0.764 (0.143)
Median (IQR)	0.0683 (0.440)	0.742 (0.201)	0.778 (0.190)
Cohens d_s_	_	0.50 ^a^	0.74 ^a^
EQ-VAS (%) ¶: *n* (% of total)	33 (94.3%)	20 (95.2%)	13 (100%)
Mean (SD)	62.6 (21.7)	72.0 (17.8)	65.8 (23.8)
Median (IQR)	60.0 (30.0)	75.0 (22.5)	70.0 (35.0)
Cohens d_s_	-	0.46 ^a^	0.14
Exercise Self Efficacy (score) ¶: *n* (% of total)	35 (100%)	16 (76.2%)	8 (61.5%)
Mean (SD)	18.9 (4.1)	21.3 (5.3)	21.8 (6.1)
Median (IQR)	18.0 (5.0)	19.5 (9.3)	23.0 (9.8)
Cohens d_s_	-	0.54 ^b^	0.63 ^b^

Key: ¶ higher score is better; $ lower score is better. SD Standard deviation; IQR Interquartile range; Cohens d_s_ effect sizes—^a^ small 0.2–0.5, ^b^ medium 0.5–0.8, ^c^ large 0.8–1.

## Data Availability

Data are available upon request from the corresponding author F.H. at fhorgan@rcsi.ie.

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
