# Peer review of "Enhancing Existing Formal Home Care to Improve and Maintain Functional Status in Older Adults: Results of a Feasibility Study on the Implementation of Care to Move (CTM) in an Irish Healthcare Setting"

_ijerph, 2022, doi:10.3390/ijerph191811148_

Round 1

Reviewer 1 Report

It was a good opportunity to review interesting research topics.

 However, since Ireland's health care system is different from other countries, there are doubts about how useful the results of this study will be.
However, the paper itself is well written, and there are no major problems with the research methodology, so I am going to accept this research paper.

I just want to recommend two things.

1) No matter how much protocol has been published, I would like you to explain the CTM approach in detail.
The important thing is to explain how you proceeded to embed CTM in home care.
What is "a series of consistent movement prompts"?

2. Please also add a limitation for the part monitored by telephone due to the COVID-19 pandemic.

Author Response

We are very grateful to all the reviewers for their very positive feedback. We have responded to the reviewer comments and have revised the manuscript.

Detailed response are included here:

Reviewer 1:

Comment: It was a good opportunity to review interesting research topics.

However, since Ireland's health care system is different from other countries, there are doubts about how useful the results of this study will be.

Reply: Thank you for the comment. However, although Ireland's health care system is different in regards of funding, the value of supporting movement in healthcare interactions supporting people to live in their own home, we feel the results of this study have a place in all countries that have a home care system in place, private or otherwise.

Comment: However, the paper itself is well written, and there are no major problems with the research methodology, so I am going to accept this research paper.

Reply: Thank you

Comment: I just want to recommend two things.

1) No matter how much protocol has been published, I would like you to explain the CTM approach in detail.

The important thing is to explain how you proceeded to embed CTM in home care.

What is "a series of consistent movement prompts"?

Reply: We have added a slightly longer explanation of the CTM approach, however, the original protocol and TIDIER checklist with very detailed explanations is available open access so we have kept our response relatively brief. We have added “The CTM training for support workers has three key themes/approaches to allow this ‘embedding’; Communication skills to have purposeful conversations about movement (providing a structured framework); A series of targeted, specific movements and prompts (for key movements already being performed as part of the usual package of care, daily living); Where applicable, motivating and empowering older people to carry out home exercise

programmes prescribed by therapy services. The specific movements and prompts included ‘prepare to move’ prompts (hip and buttock movement towards the front of the chair before a rise, foot placement and powering upright with some foot pedals to aid circulation and reduce chances of postural hypotension); prompts to improve ‘ADLs’ (heel raises and knee bends to get things in/out of cupboards, balance tasks waiting for kettle to boil near a solid fixed support); reviewing successes and movements since last visit or during visit. ”  On page 3, line 133 – page 4, line 144, before the reference to the published detailed intervention description sentence.

Comment:

2) Please also add a limitation for the part monitored by telephone due to the COVID-19 pandemic.

Reply: We have added a one sentence limitation to the manuscript – ‘Some of the outcome measures had to be gathered by telephone rather than face to face due to social restrictions in place during the pandemic. It is possible that this led to some outcome measures being less accurately reported” on page 19, lines 624-627 of manuscript.

Reviewer 2 Report

The manuscript entitled “Enhancing existing formal home care to improve and maintain functional status in older adults: Results of a feasibility study on the implementation of Care to Move (CTM) in an Irish healthcare setting” evaluates the new approach in older adults receiving home care in pre-and post-the Covid-19 pandemic. The article is clearly presented with the focus on the above mentioned.  Overall this is a well presented manuscript very much relevant to the literature required in this field.

The manuscript is well written, and the authors have deeply investigated the home care in older adults. Moreover, according to me, the manuscript sends out a very important message to the physiotherapy community, i.e., an urgent need to home support for patients in the Covid-19 pandemic and in a post-pandemic world.

Author Response

We are very grateful to all the reviewers for their very positive feedback. We have responded to the reviewer comments and have revised the manuscript.

Reviewer 2:

Comment: The manuscript entitled “Enhancing existing formal home care to improve and maintain functional status in older adults: Results of a feasibility study on the implementation of Care to Move (CTM) in an Irish healthcare setting” evaluates the new approach in older adults receiving home care in pre-and post-the Covid-19 pandemic. The article is clearly presented with the focus on the above mentioned.  Overall this is a well presented manuscript very much relevant to the literature required in this field.

The manuscript is well written, and the authors have deeply investigated the home care in older adults. Moreover, according to me, the manuscript sends out a very important message to the physiotherapy community, i.e., an urgent need to home support for patients in the Covid-19 pandemic and in a post-pandemic world.

Reply: Thank you for your positive comments.

Reviewer 3:

Comment: congratulations for the very interesting article

Reply: Thank you for your positive comment.

Note to Editor: This reviewer ticked that extensive editing of English language and style required, but the other two reviewers have not. The authors are English speaking as native language so we feel this may have been ticked inadvertently.

Reply: Nevertheless, we have given the manuscript a careful re-read to remove potential language errors and inconsistencies, and we added a number of commas and hyphens to aid readability of the manuscript. All of the changes are highlighted in yellow text highlight.

Reviewer 3 Report

congratulations for the very interesting article

Author Response

Note to Editor: This reviewer ticked that extensive editing of English language and style required, but the other two reviewers have not. The authors are English speaking as native language so we feel this may have been ticked inadvertently.

Reply: Nevertheless, we have given the manuscript a careful re-read to remove potential language errors and inconsistencies, and we added a number of commas and hyphens to aid readability of the manuscript. All of the changes are highlighted in yellow text highlight.

Author email / affiliation correction request

We also wish to correct the email for Prof. Maria O’Sullivan please – it should be as follows page 1 line 16

[email protected]

We have corrected the affiliation for Prof Rose Galvin and Dr Elissa Burton.

All of the changes are highlighted in yellow text highlight.